# Topography and soil variables drive the plant community distribution pattern and species richness in the Arjo-Diga forest in western Ethiopia

**Tariku Berihun Tenaw**[1,2*], **Tamrat Bekele Gode**[2], **Ermias Lulekal Molla**[2], **Zemede Asfaw Woldemariam**[2]

**1** Department of Biology, Dilla University, Dilla, Ethiopia, **2** Department of Plant Biology and Biodiversity Management, Addis Ababa University, Addis Ababa, Ethiopia

☯ These authors contributed equally to this work.
* berihun.tariku@yahoo.com

**Data Availability Statement:** All relevant data are within the paper and its Supporting Information files.

**Funding:** The author(s) received no specific funding for this work.

## Abstract

Understanding plant community characteristics, distributions, and environmental relationships is crucial for sustainable forest management. Thus, this study examined the relationships between plant community composition and topographic and soil variables within the Arjo-Diga forest. Vegetation data were collected from 72 nested plots (30 × 30 m² and 2 × 2 m²) systematically laid along nine transects spaced 300 to 700 m apart. Environmental variables, including soil properties and anthropogenic disturbance, were recorded within each main plot. Agglomerative hierarchical cluster analysis and canonical correspondence analysis (CCA) using R software were employed to identify distinct plant community types and examine their relationships with environmental factors. The Shannon–Wiener diversity index was calculated to quantify and compare species diversity among the identified community types. The analysis revealed five distinct plant community types: 1: *Maesa lanceolata-Ehretia cymosa*, 2: *Trichilia dregeana-Flacourtia indica*, 3: *Acacia abyssinica-Millettia ferruginea*, 4: *Combretum collinum-Croton macrostachyus*, and 5: *Terminalia macroptera-Piliostigma thonningii*. The CCA results highlighted the significant influence ($p < 0.05$) of altitude, CEC, TN, and disturbance on species distribution and plant community formation. The findings indicate that variation in plant communities is closely associated with altitude, TN, and CEC, as well as with disturbance factors such as human interventions, with elevation being the most influential factor. Based on these findings, it is recommended that conservation plans consider the effects of human interventions to address the challenges in conserving forests in the future. Additionally, further research efforts should focus on mitigating disturbance factors and understanding the environmental variables that affect forests to improve their protection.

**Competing interests:** The authors have declared that no competing interests exist.

## Introduction

The relationship between biodiversity and ecosystem function is a fundamental area of study in ecology [1]. Thus, investigating the relationships among vegetation attributes, such as richness, diversity, evenness, and environmental factors, is vital for developing effective conservation strategies [2]. The interactions among various environmental factors can lead to changes in habitat conditions, directly or indirectly affecting species distribution and plant diversity worldwide. Plant species composition, diversity, and spatial distribution patterns are influenced by both abiotic and biotic factors [3]. Understanding the complex relationship between plants and abiotic factors is essential, and further research into their influence on species distribution and the formation of plant communities is crucial [4].

Most studies of environmental factors have focused on elevation, topography, and soil to study how environmental factors affect species composition and plant diversity in different ecosystems. In particular, altitude drastically alters abiotic factors such as water, temperature, and soil composition, which directly affect plant growth and development [5,6]. On a global scale, altitude also regulates the response of plant communities to environmental factors [7]. A study by Pandey et al. [8] reported different patterns of species richness along different elevation gradients. Several studies have shown that species richness decreases monotonically from lowest to highest elevations [9,10]. Hump-shaped patterns have also been reported at mid-elevations [11]. However, some researchers have shown low species richness at mid-elevations [12]. These contrasting results suggest species richness along elevation gradients is not a general trend since many species exhibit different phenotypic traits, such as leaf characteristics, biomass, and phenology. As a result, species richness patterns and plant species distributions along altitudinal gradients often differ meaningfully among conservationists.

Soil is another important environmental variable that shapes plant diversity and vegetation patterns by forming diverse habitats. Deficiencies in soil nutrients have been reported to impact various aspects of forests in tropical forests, including community structure, plant biomass, tree height, and basal area [13]. Conversely, forests with high species richness often exhibit high nutrient dynamics in specific locations. The influence of soil factors, such as total nitrogen (TN) and available phosphorus, on plant community structure is well established [14]. In particular, total nitrogen (TN) is a limiting factor for plant growth [15] and significantly influences plant diversity and community composition.

Additionally, studies have demonstrated that species richness can be influenced by high nitrogen deposition [16]. In tropical forests, the soil cation exchange capacity (CEC) has been found to impact tree species richness [17,18]. However, knowledge gaps must be addressed to understand how soil cation exchange capacity (CEC) affects tree species richness in tropical forests.

Species composition, diversity, and distribution are also influenced by anthropogenic disturbances such as agricultural expansion, settlement, livestock overgrazing, selective cutting, and fires [19]. Disturbance positively affects vegetation properties to a certain extent [20]. Under conditions of severe disturbance, species richness and diversity are low because most species cannot tolerate frequent destructive events. However, due to dominant competitors and fast colonizers, high species richness can be predicted at moderate levels of disturbance [21]. Understanding how species respond to human-caused disturbances is vital to making informed conservation decisions. This understanding enables practical actions such as habitat conservation and restoration to maintain critical ecological phenomena such as species distribution limits.

Various authors in Ethiopia, such as Gurmessa et al. [22], Dibaba et al. [23] and Addi et al. [24], have documented the results of studies on biodiversity, structure, and regeneration status

in different moist Afromontane forests in other parts of Ethiopia. However, many of their species are threatened, endangered, or locally extinct due to habitat destruction, fragmentation, and overexploitation of forest products and habitats. Understanding the interaction between species diversity and composition and the relationships between environmental factors and plant community types remains critical for developing forest management strategies.

The Arjo-Diga forest is a Moist Afromontane Forest (MAF) and Combretum-Terminalia woodland (CTW) vegetation type located in western Ethiopia [25]. However, the relationships between various environmental factors, including cation exchange capacity, total nitrogen, soil organic carbon, phosphorus, disturbances, altitude, slope, and plant species distribution, in forest ecosystems have not been studied. To overcome this scientific data gap, investigating plant species diversity, community type richness, and community distribution patterns along environmental gradients is crucial for ensuring effective and sustainable management of the Arjo-Diga forest. Therefore, the present study aimed to (1) examine floristic composition and species diversity and (2) assess the relationships between plant and environmental variables in the study area.

## Materials and methods

### Description of the study area

The study was conducted in the Arjo-Diga forest, located in the Diga District, Oromia Regional State, in the southeastern part of Ethiopia. It lies at an elevation between 1,200 and 2,220 m.a.s.l. and covers an area of approximately 12,683.6 hectares. Nekemte is a major town nearby 340 km away from Addis Ababa, the capital of Ethiopia. The Arjo-Diga forest extends between 9˚59′00″ N and 9˚6′30″ N and 36˚18′30″E and 36˚24′30″ E (Fig 1). The forest is bounded by three districts: Guto Gida, Chewaka, Sasiga, and Leka Dulecha. The topography of

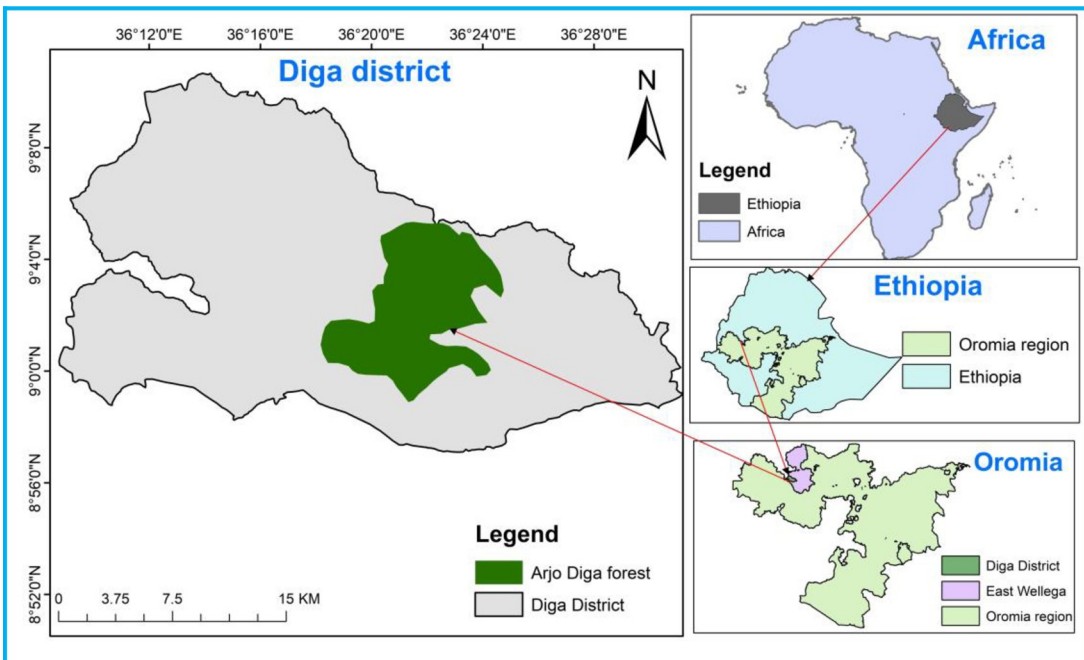

**Fig 1. Map of Africa showing Ethiopia, the Oromia region, and the study district (source: OCHA, 2021) (https://data.humdata.org/dataset/cod-ab-eth).**

the study area varies, ranging from flat to gentle slopes to moderate to steep slopes. The slope gradients range from 7% to 30% in the southwest and 30% to 60% in the northeast, increasing the area's susceptibility to severe erosion. The study area comprises two agroecological zones: the lowlands (51.4%) and the midlands (48.6%) [26]. Steep slopes characterize the midlands and are predominantly covered by forests.

## Soil and geology

Acriosols are the dominant soil type in the study area and are characterized by heavy weathering and acidity. These soils exhibit a low cation exchange capacity ranging from 15 to 25 cmolc kg$^{-1}$, a pH of 5.2, and a deficiency in phosphorus content [27]. The predominant soil colors observed were red in the midlands and black in the lowlands. Geologically, the study area comprises Precambrian rocks of high origin, including migmatites [28].

## Climate

The study area mainly experiences tepid to cool subhumid mid-highlands in the northeastern part and hot to warm humid lowlands in the northwestern part. The maximum temperature was 25.7°C, and the minimum was 0.2°C (Fig 2). The mean annual precipitation is approximately 1977 mm/year, with an unimodal precipitation pattern. The monthly mean precipitation trend shows its maximum in July, June, and August. It is dry from November and extends into January.

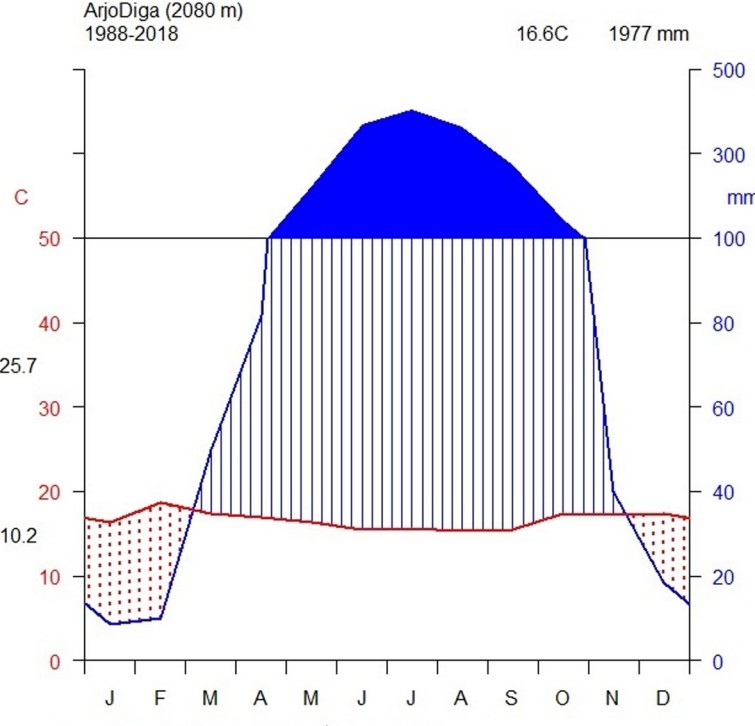

**Fig 2. Climate diagram of the study area showing the rainfall distribution and temperature variation from 1988–2018 (Data Source: National Meteorological Agency, 2020).**

## Vegetation types

Based on the most recent improved classification of the potential vegetation of Ethiopia, the Arjo-Diga forest belongs to the Moist Afromontane forest vegetation types [29]. The vegetation is distinguished from others by the presence of emergent species, including *Pouteria adolf-friederici* (Engl) Baehni. Other characteristic species in this forest include *Sapium ellipticum* (Hochst) and Pax *Celtis africana* Burm.f. *Euphorbia ampliphylla* Pax., *Ficus sur* Forssk, *Schefflera abyssinica* (Hochst. ex Rich.) Harms, *Syzygium guineense ssp. afromontanum* F. White, *Nuxia congesta* R.Br. ex Fresen, *Galiniera saxifraga* (Hochst.) Bridson, *Rytigynia neglecta* (Hiem) Robyns, *Vepris dainellii* (Pichi-Serm)Kokwaro, *Rothmannia urcelliformis* (Hiem) Bullock ex.Robym, and several *Albizia* species. Dominant woody climbers in the forest include *Combretum paniculatum* Vent *Landolphia buchananii* (Hallier f.) Stapf and *Urera hypselodendron* (Hochst. ex A.Rich.) Wedd. The western part of the study area borders the Didessa Valley, characterized by *Combretum-Terminalia* vegetation.

## Methods of data collection

**Reconnaissance survey and sampling technique.** A reconnaissance survey was conducted in October 2017 in the Arjo-Diga forest to get an impression of the site conditions and identify the sampling sites in the study area. The actual fieldwork was conducted between November 2016 and January 2017. A systematic sampling technique was employed for vegetation and environmental data collection to ensure complete coverage of ecological variation and habitat heterogeneity. Seventy-two sample plots along nine-line transects were established using a Garmin GPS H72. The distances between each transect ranged from 300 to 700 m apart, and the sampling plots were 200 m apart. A square plot of $30 \times 30$ m (900 m$^2$) was used to collect data on woody species. Five $2 \times 2$ m (4 m$^2$) subplots (4 at each corner and 1 in the center) were nested in each 900 m$^2$ main plot to collect data on herbaceous species.

**Vegetation data collection.** In each main plot, all individual trees and shrubs with a diameter at breast height (DBH) $\geq 2$ cm and a height $\geq 1.5$ m were measured using a caliper and clinometer, respectively. The percent cover-abundance of trees, shrubs, and lianas was visually estimated using the scale provided by Mueller-Dombois and Ellenberg [30]. Similarly, the percent cover of herbaceous species was estimated within smaller plots nested within the larger plots. Plant specimens were collected from each plot, coded, pressed, and dried. All collected voucher specimens were identified using the Flora of Ethiopia and Eritrea (volumes 1–8) and deposited in the National Herbarium of Ethiopia (ETH) at Addis Ababa University.

**Environmental data collection.** Environmental variables such as altitude, slope, and geographic coordinates were measured for each plot using a Garmin GPS H72. Soil samples were taken from the five 4 m$^2$ subplots (4 at the corners and one at the center of each main plot) using a soil auger to a depth of 30 cm. These soil samples were mixed, and a composite sample (one kg) from each main plot was taken to the laboratory for analysis. Composite soil samples were air-dried, crushed, and sieved using 2 mm sieves. The chemical properties of the soil samples were analyzed at the Bedele soil laboratory following standard analytical procedures [31]. Soil organic carbon was determined using the Walkely and Black [32] methods, total nitrogen was determined using the Kjeldahl [33] method, pH was measured using a pH meter [34], available phosphorus was measured using the Bray-I methods [35], and cation exchange capacity was determined using the ammonium acetate method [36].

Disturbance was recorded as present or absent in each sampled plot within the study area. The magnitude of disturbance in each sampled plot was rated on a scale from 0 to 3 based on visible signs of vegetation disturbance parameters. These parameters included tree cutting, firewood collection, charcoal production, debarking, grazing, forest fires, the presence of bee

hives, and the establishment of footpath signs following the procedure outlined by Hadera [37], Yeshitla and Bekele [38] and Senbeta et al. [39]. The disturbance level was coded as 0: no disturbance. 1: If any one of the disturbances mentioned above existed, albeit to a small degree (slightly disturbed); 2: if any two disturbance factors were noted (moderately disturbed); 3: denoting a significant level of human disturbance if three or more disturbance elements were present (highly disturbed).

## Data analysis

**Plant community classification.** The vegetation in the study area was classified into different community types using agglomerative clustering analysis, employing a similarity ratio as the resemblance index and the Ward method as the classification method using R software version 4.2.2 [40]. The resulting community types were refined in a synoptic table, which summarized species occurrences as synoptic cover-abundance values [41]. These synoptic values represent the product of species frequency and average cover-abundance. Finally, the plant community types were named after the synoptic values of two dominant species with an indicator value of $p < 0.05$.

Canonical correspondence analysis (CCA) was employed to establish correlations between the identified plant community types and selected environmental and disturbance factors. The decision to use CCA ordination was based on the observation that the first axis in the detrended correspondence analysis (DCA) had a value greater than 4 (specifically 5.6), indicating the presence of heterogeneous environmental datasets in the study [42]. The CCA analysis examined various environmental factors, including altitude, slope, soil chemical properties (pH, available phosphorus, soil organic carbon, total nitrogen, and cation exchange capacity), and disturbance factors, such as tree cutting, debarking, grazing, fire, timber extraction, and charcoal production. The associations between these factors and the identified plant community types were investigated. Additionally, one-way ANOVA followed by post hoc Tukey HSD tests was used to determine whether there were significant differences in the mean environmental variables, species richness, diversity, and evenness among the plant communities. Pearson correlation analysis was also employed to verify the linear relationships among the explanatory variables, such as soil chemical properties, disturbances, and topographic variables.

**Community diversity analysis.** The Shannon Weiner diversity index (H'), species richness, and Shannon evenness (J) were calculated to describe community diversity using R software version 4.2.2 [40].

$$H' = -\sum_{i=1}^{s} \text{Pi ln Pi} \qquad (1)$$

where H' is the Shannon–Weiner diversity index, s is the number of species, and pi is the proportion of the ith species. ln = the natural logarithm.

Shannon's evenness index (J) was calculated by using the following equation:

$$J = \frac{H'}{H'\text{max}} \qquad (2)$$

where $H'$ is the Shannon–Wiener diversity index and Hmax = lns, where s is the number of species in the plot.

Sorensen's similarity coefficient (Ss) was used to compare the similarity between two communities [43].

$$Ss = \frac{2a}{(2a + b + c)} \tag{3}$$

where 'a' represents the number of species present in both communities, 'b' represents the number of species unique to the first community, and 'c' represents the number of species unique to the second community.

## Results

### Floristic composition

This study identified 234 plant species distributed across 183 genera and 73 families. The complete list of these species can be found in (S1 Table). Among the families, those with the highest species richness were Asteraceae (29 species), Fabaceae (23 species), Euphorbiaceae (14 species), Lamiaceae (12 species), and Rubiaceae (11 species). On the other hand, eight families had a species count ranging from four to six (Table 1). The recorded plant species were also classified into different growth forms, including trees, shrubs, herbs, and climbers. Trees accounted for 61 species (26%), shrubs comprised 70 species (30%), herbs constituted 84 species (36%), and climbers represented 19 species (8%) (Fig 3).

Of the total identified plant species, 20 (8.5%) were endemic to the flora of Ethiopia and Eritrea (S1 Table). Among these endemic species, *Crotalaria rosenii*, *Impatiens tinctoria*, *Kalanchoe petitiana*, *Pycnostachys abyssinica*, and *Solanecio gigas* are newly documented species that have not been previously reported in the floristic region of Wollega (Table 2).

### Plant community types

This study employed hierarchical cluster analysis to identify five distinct plant community types (Fig 4). The dendrogram was constructed using the similarity ratio and Ward's method, illustrating the relationships between the five clusters based on percent species coverage data. The analysis involved clustering a data matrix consisting of 72 plots and 234 plant species using agglomerative hierarchical cluster analysis. Each community type was named after one or two dominant indicator tree and shrub species, chosen based on their significant indicator

**Table 1. Dominant families in the study forest accounting for many genera and species.**

| Family | Number of genera | Number of species |
|---|---|---|
| Asteraceae | 17(10%) | 29(12.4%) |
| Fabaceae | 16(8.7%) | 23(9.8%) |
| Rubiaceae | 11(7%) | 11 (4.7%) |
| Euphorbiaceae | 10(5.5%) | 14 (6%) |
| Lamiaceae | 10(8.7%) | 12(5.1%) |
| Acanthaceae | 6(2.9%) | 6(2.6%) |
| Poaceae | 6(2.9%) | 6(2.6%) |
| Malvaceae | 4(2.2%) | 5(2.1%) |
| Rhamnaceae | 4(2.2%) | 5(2.1%) |
| Rutaceae | 4(2.2%) | 4(1.7%) |
| Sapindaceae | 4(2.2%) | 6(2.6%) |
| Verbenaceae | 4(2.2%) | 4(1.7%) |
| Vitaceae | 4(2.2%) | 5(2.1%) |

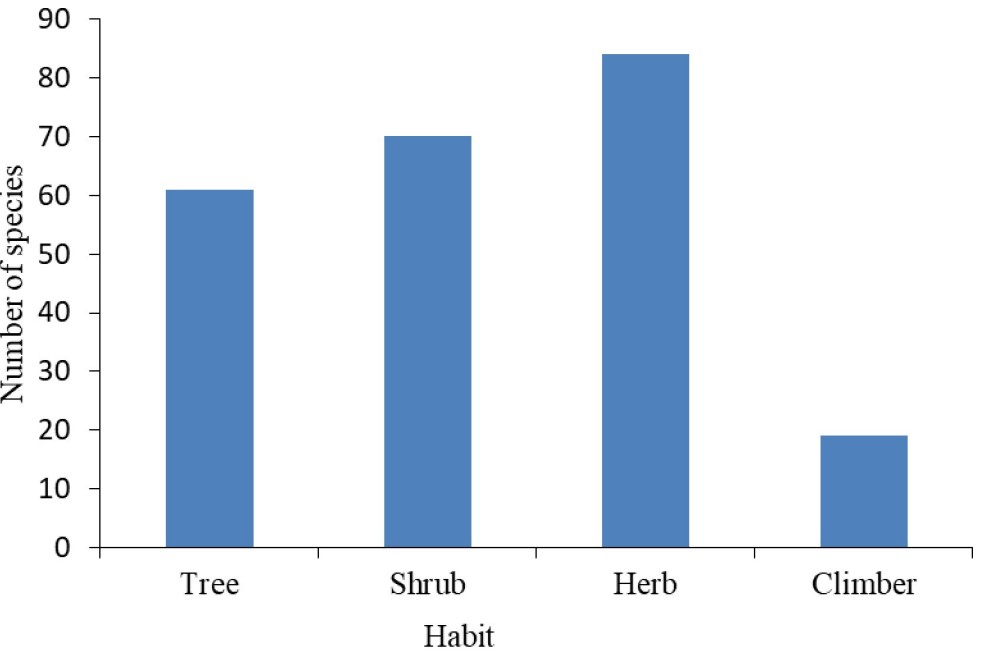

**Fig 3. Habits of plant species in the study area.**

values and synoptic mean abundances (Table 3). In this study, a species was considered an indicator of a particular group if its indicator value was statistically significant ($p < 0.05$) (S2 Table). Consequently, five plant communities types were identified in this study: *Maesa lanceolata-Ehretia cymosa*, *Trichilia dregeana-Ficus vasta*, *Acacia abyssinica-Millettia ferruginea*, *Combretum collinum-Bersama abyssinica*, and *Terminalia macroptera-Piliostigma thonningii*. Each community type is described as follows:

**Table 2. Newly documented species and endemic species identified from the Wollega floristic region.**

| Species name | Family | Habit | Species distribution in the Flora Region |
|---|---|---|---|
| *Crepis foetida* | Asteraceae | H | EW, TU, GD,SU,AR, HA |
| *Crotalaria rosenii** | Fabaceae | S | SU, AR, BA, KF,SD |
| *Cyperus distans* | Cyperaceae | H | AF, GD,SU,IL,KF,GG, SD |
| *Euphorbia ampliphylla* | Euphorbiaceae | T | TU, GD, GJ, WU, SU, IL, KF, SD, HA |
| *Impatiens tinctoria** | Balsaminaceae | H | GD, SU, AR, KF, SD, BA |
| *Kalanchoe petitiana** | Crassulaceae | H | EW, GD,WU,SU,AR,BA, HA |
| *Maytenus arbutifolia* | Celastraceae | T | SU, AR, SD,GG |
| *Olea capensis subsp. Macrocarpa* | Oleaceae | T | GD, SU, AR, IL, KF, SD, BA |
| *Pycnostachys abyssinica** | Lamiaceae | S | IL, KF, GG, SD |
| *Solanecio gigas** | Asteraceae | S | GD, GJ, WU, SU, AR, SD, IL, KF, BA |
| *Solanum incanum* | Solanaceae | S | EW, EE, TU, SU, GG, SD, BA, AR |
| *Tagetus minuta* | Asteraceae | H | EW, TU, GD, WU, SU, KF, SD, BA, HA |
| *Trifolium semiplosum* | Fabaceae | H | EW, TU, GD, WU, SU, AR, BA, SD |

Note: Habit (T = Tree, S = Shrub, H = Herb); EW = Eritrea west, Eretria, TU = Tigray region, Ethiopia, GD = Gondar region, Ethiopia, SU = Shewa upland, Ethiopia, AR = Arsi region, Ethiopia, WL = Wellega region, Ethiopia, IL = Ilubabor region, Ethiopia, KF = Kefa region, Ethiopia, GG = Garno Gofa region, Ethiopia, SD = Sidamo region, Ethiopia, BA = Bale region, Ethiopia, HA = Hararge region, Ethiopia, WU = Welo region, Ethiopia, EE = Eritrea East, and * stands for newly documented endemic species in the Wollega floristic region.

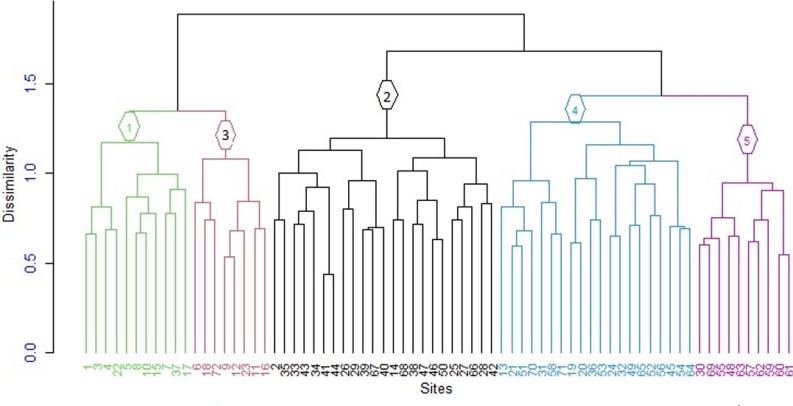

**Fig 4. Dendrogram showing the five distinct plant community types, 1 (Community type 1) =** *Maesa lanceolata-Ehretia cymosa* **community type, 2 (Community type 2) =** *Trichilia dregeana–Ficus vasta* **community type, 3 (Community type 3) =** *Acacia abyssinica-Millettia ferruginea* **community type, 4 (Community type 4) =** *Combertum collinum–Bersama abyssinica* **community type, and 5 (Community type 5) =** *Terminalia macroptera-Piliostigma thonninigii* **community type.**

*Maesa lanceolata-Ehretia cymosa* community type (C1)

This community type is situated at the highest elevation within the study area, from 1608 to 1992 m.a.s.l. It comprises 110 species. Among the observed tree species, notable dominants included *Albizia gummifera*, *Celtis africana*, *Millettia ferruginea*, and *Prunus africana*. In the herbaceous layer, the dominant species consisted of *Agratum conzoides*, *Bidens pilosa*, *Achyranthus aspra*, and *Digitaria abyssinica*. Common lianas/climbers found in this community include *Clematis simensis*, *Dioscorea praehensilis*, *Phytolaca decandra*, *Rhamnus prinoides*, *Landolphia buchananii*, *Cyphostemma cyphopetalum*, and *Urera hypselodendron*. The community exhibited five indicator species with significant indicator values ($p < 0.05$) (S2 Table). Additionally, this community is characterized by economically important species such as *Coffea arabica*, *Dioscorea praehensilis*, and *Piper capense*. *Pteridium aquilinum*, a dominant fern species, occurs exclusively within this community. Each community type is described as follows:

*Trichilia dregeana–Ficus vasta* community type (C2)

This community was found between 1537 and 1631 m.a.s.l. and included 23 plots and 148 species. *Syzygium guineense subsp. guineense*, *Ficus vasta*, and *Croton macrostachyus* were the dominant tree species, along with *Trichilia dregeana* and *Flacourtia indica*. Other associated tree species are *Apodytes dimidiata*, *Combertum collinum*, *Ficus mucuso*, *Warburgia ugandensis*, and *Cordia africana*. There were three indicator species, namely, *Cyathula cylindrical*, *Flacourtia indica*, and *Bersama abyssinica*, with significant indicator values ($p<0.05$) (S2 Table). *Maytenus gracilipes* and *Carisa spinarum* were the dominant shrub species. The herb layer was dominated by *Oplismenus hirtellus*, *Laggera crispata*, and *Kalanchoe petitiana*. Lianas in this community include *Jasminum abyssinicum*, *Saba comonesis*, *Capparis tomentosa*, *Smilax anceps*, *Rubus apetalus*, and *Combretum paniculatum*.

*Acacia abyssinica-Millettia ferruginea* community type (C3)

This community is spread across altitudes from 1476 to 1792 m.a.s.l. and includes eight plots and 94 species. The dominant tree species in the community are *Croton macrostachyus*, *Albizia gummifera*, *Buddleja polystachya*, *Acacia abyssinica*, and *Millettia ferruginea*. Other prominent tree species include *Combertum collinum*, *Cordia africana*, *Bridelia micrantha*,

**Table 3. Synoptic mean cover value of species in each community type.**

| Species name | Community types | | | | |
|---|---|---|---|---|---|
| | C1 | C 2 | C 3 | C 4 | C 5 |
| *Maesa lanceolata* | **2.73** | 1.87 | 1.37 | 0.85 | 1.8 |
| *Ehretia cymosa* | **2.32** | 0 | 0 | 0.1 | 0 |
| *Albizia gummifera* | 1.73 | 0.56 | 1.75 | 1.5 | 0 |
| *Cordia africana* | 1.17 | 1.13 | 1.33 | 0.55 | 1.32 |
| *Trichilia dregeana* | 1.16 | **4.33** | 1.08 | 1 | 3 |
| *Flacoutia indica* | 1.03 | 2.18 | 2.94 | 2.6 | 2.8 |
| *Ficus exasperate* | 0.11 | 2.9 | 1.78 | 2 | 1.98 |
| *Ficus vasta* | 0.54 | **3.18** | 0.65 | 0.25 | 1.45 |
| *Acaccia abyssinica* | 0.64 | 1.6 | **5.31** | 0.1 | 0 |
| *Millettia ferruginea* | 0.09 | 1.34 | **3.35** | 0.15 | 1 |
| *Stereospermum kunthianum* | 0 | 1.3 | 2.33 | 1.75 | 3 |
| *Apodytes dimidiate* | 0 | 1.23 | 2.24 | 1.77 | 0.5 |
| *Ficus thonningii* | 1.09 | 0 | 1.67 | 0.5 | 0.3 |
| *Maytenus gracilipes* | 0.63 | 1.34 | 1.37 | 0.15 | 0 |
| *Combretum collinum* | 0.18 | 2.43 | 0 | **4.54** | 0.6 |
| *Croton macrostachyus* | 0.17 | 1.91 | 0.34 | 2.31 | 3.1 |
| *Bersama abyssinica* | 1.63 | 1.86 | 0.62 | **2.65** | 0.9 |
| *Combretum molle* | 0 | 0.08 | 0 | 2.01 | 1.3 |
| *Bridelia macrantha* | 0.09 | 1.73 | 1.67 | 1.65 | 0.9 |
| *Terminalia macroptera* | 0 | 0.56 | 3.37 | 1.65 | **4.42** |
| *Piliostigma thonningi* | 0.09 | 0 | 2 | 0.45 | **3.34** |
| *Hymenodictylon floribundum* | 0 | 0 | 0.12 | 0.2 | 2.13 |
| *Ximenia americana* | 0 | 0.73 | 1.3 | 2.1 | 1 |
| *Warburgia ugandensis* | 0.09 | 1.17 | 0.65 | 1.54 | 0.4 |
| Number of plots | 11 | 23 | 8 | 20 | 10 |

The **bold** values indicate the species used to name each plant community type.

*Croton macrostachyus*, and *Maesa lanceolata*. The shrub layer is dominated by species such as *Maytenus gracilipes*, *Senna petersiana*, *Grewia ferruginea*, *Vernonia auriculifera*, *Brucea antidysenterica*, and *Nuxia congesta*. Dominant species, including *Sonchus bipontini*, *Tristemma mauritianum*, *Achyranthus aspra*, *Bidens macroptera*, and *Oplismenus hirtellus* characterize the herb layer. Regarding lianas and climbers, the community is characterized by *Paullinia pinnata* and *Cyphostemma adenocaule*. Additionally, this community exhibited seven indicator species with significant indicator values (p < 0.05) (S2 Table).

*Combertum collinum– Bersama abyssinica* community type (C4)

This community spread at altitudes between 1630–1711 m.a.s.l. *Pouteria adolfi-friederici* was the only emergent tree species found in this community. *Stereospermum kunthianum*, *Terminalia macroptera*, and *Bridelia micrantha* were the dominant tree species, followed by *Combertum collinum* and *Croton macrostachyus*. Other tree species are *Ficus sycomorus*, *Entada abyssinica*, *Ximenia americana*, *Albiza gummifera*, and *Flueggea virosa*. The shrub species are dominated by *Clausena anisata*, *Vepris dainellii*, *Gardenia ternifolia*, and *Acanthus polystachius*. Dominant herb layer species are *Haumaniastrum villosum*, *Helinus mystacinus*, *Indigofera spicata*, *Senna occidentalis*, *Pennisetum thumbergi*, and *Crotalaria lachnophora*. *Clematis longicauda* and *Clematis simensis* are the most common climbers in this community. This

community has three indicator species, namely, *Combretum collinum*, *Maytenus obscura*, *Gnidia glauca*, and *Guizotia villosa*, with significant indicator values (p < 0.05) (S2 Table).

*Terminalia macroptera-Piliostigma thonninigii* community type (C5)

This community is represented by ten plots and 98 species, located within an altitude range of 1501 to 1672 m.a.s.l. The dominant tree species in this community include *Stereospermum kunthianum and Syzygium guineense subsp. afromontanum*, *Ficus exasperata*, *Gardenia ternifolia*, and *Flueggea virosa*. Other prominent tree and shrub species found in the community include *Combretum molle*, *Ficus mucuso*, *Albizia malacophylla*, *Pavetta abyssinica*, *Maytenus obscura*, *Ximenia americana*, *Oxythenteria abyssinica*, and *Flacourtia indica*. The herb layer is composed of species *Piliostigma thonningii*, such as *Crotalaria ononoides*, *Aframomum alboviolaceum*, *Conyza sumatreosis*, *Alysicarpus rugosus*, and *Guizotia villosa*. Among the climbers/vines present in this community, *Ampelocissus schimperiana and Cissampelos pareira* are the most common. Additionally, several indicator species with significant indicator values have been identified within this community. *Gardenia ternifolia*, Syzygium guineense subsp.*afromontanum*, and *Terminalia macroptera* are most important, as indicated in Table 3.

## Plant community composition and similarity among the five community types

Computation of the Shannon–Wiener diversity index revealed that Community 4 exhibited the highest species diversity and richness among the studied community types, followed by Community 2 and Community 1 (Table 4). However, when considering evenness, the order of the communities with decreasing evenness was 3, 5, and 1. This order does not align with the arrangement of the communities based on decreasing species richness.

Based on Sorensen's similarity coefficient, Communities 2 and 4 significantly overlapped in species composition, with 79% of the species being shared between them (Table 5). Similarly, Communities 1 and 2 also demonstrated relatively high similarity, indicating considerable species overlap. However, the similarity between Communities 3 and 5 is comparatively lower. Notably, this difference in similarity can be attributed to various environmental factors, including altitude, anthropogenic influences, and soil type, all of which were considered in the present study.

## Plant community along environmental variables

In this study, we employed canonical correspondence analysis (CCA) to investigate the distribution of plant community types along environmental variables based on the detrended correspondence analysis (DCA) results. Through the use of a free Monte Carlo test (with the Adonis function), we found that among the initial set of eight environmental variables, altitude, slope, CEC, disturbances, pH, and TN had significant influences (p < 0.05) on the community distribution (**Fig 5** and **Table 6**). Specifically, communities 1, 2, 4, and 5 were strongly

**Table 4. Diversity, richness, and evenness of plant communities.**

| Community types | Species richness | Shannon diversity (H') | Shannon- Evenness (J) |
|---|---|---|---|
| Community 1 | 110 | 4.1 | 0.87 |
| Community 2 | 148 | 4.3 | 0.86 |
| Community 3 | 94 | 3.9 | 0.89 |
| Community 4 | 162 | 4.4 | 0.85 |
| Community 5 | 98 | 4 | 0.88 |

**Table 5. Similarity indices among community types in the study area.**

| Community types | CI | C2 | C3 | C4 | C5 |
|---|---|---|---|---|---|
| C1 | 1 | | | | |
| C2 | 0.68 | 1 | | | |
| C3 | 0.34 | 0.36 | 1 | | |
| C4 | 0.45 | 0.79 | 0.39 | 1 | |
| C5 | 0.35 | 0.42 | 0.22 | 0.57 | 1 |

C1 = Community type 1, C2 = Community type 2, C3 = Community type 3, C4 = Community type 4, and
C5 = Community type 5.

associated with altitude, while community 3 had associations with CEC and TN. Moreover, communities 4, 5, and 2 were strongly influenced by disturbance.

Among the constraining environmental variables, altitude had the highest biplot score (0.95) on the first CCA axis, followed by TN, with a biplot score of 0.44 (Table 7). In contrast, the disturbance had the highest biplot score (0.94) on the second axis (CCA2), followed by CEC, with a biplot score of 0.42. All significant environmental variables, except pH and

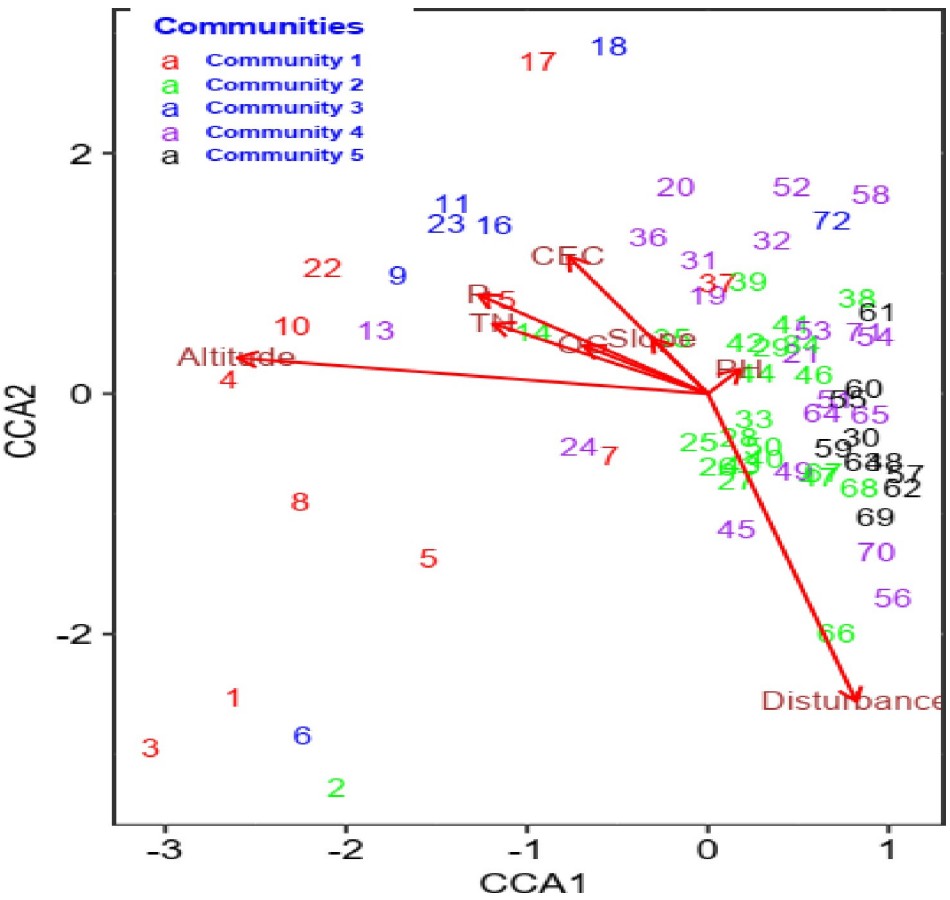

**Fig 5. The CCA ordination graph illustrates significant environmental variables (p<0.05) and their relationships with plant communities.** Arrows represent ecological factors, with their lengths indicating contributions to axes. The numbers correspond to the plot numbers, and the angle between the arrows and axes denotes the variable-ordination axis correlation.

**Table 6. Results of the Monte Carlo test using the Adonis functional for the environmental variables.**

| Environmental variables | Df | Sum of Squares | $R^2$ | F | Pr(>F) |
|---|---|---|---|---|---|
| Altitude | 1 | 1.2540 | 0.0452 | 3.3734 | 0.001*** |
| Slope | 1 | 0.4280 | 0.0154 | 1.1513 | 0.027* |
| Disturbance | 1 | 1.2771 | 0.0460 | 1.1452 | 0.002** |
| SOC | 1 | 0.2935 | 0.0105 | 0.7861 | 0.425 |
| TN | 1 | 0.4328 | 0.0156 | 1.1643 | 0.01* |
| **Av.** P | 1 | 0.2775 | 0.0103 | 0.8138 | 0.79 |
| pH | 1 | 0.3925 | 0.0141 | 1.0560 | 0.03* |
| CEC | 1 | 0.5130 | 0.0185 | 1.3799 | 0.005* |
| Residuals | 63 | 19.80091 | 0.8449 | | |
| Total | 71 | 27.7161 | 1.0000 | | |

*, ** and *** indicate significance at p<0.05, 0.01 and p<0.001, respectively.

SOC = soil organic carbon, TN = total nitrogen, CEC = cation exchange capacity, Av. P = available phosphorus.

disturbance, exhibited negative correlations with CCA1. Notably, altitude strongly negatively correlated with CCA1 (r = -0.95, p<0.001), followed by TN (r = -0.47, p<0.01) (Table 7). Additionally, the second axis (CCA2) demonstrated a robust negative correlation with disturbance (r = -0.94, p<0.01). The eigenvalues obtained for the first two axes were 0.34 and 0.15, respectively. The cumulative proportion of variance explained by the first six CCA axes in the joint biplot was 85%. The first and second axes significantly explain the variation in the community distribution pattern. The first axis alone accounts for a substantial proportion of the variation (30%), and the second axis also explains a notable amount (30%). Therefore, the combined effect of the first two axes explained 43% of the variation in plant community distribution and formation patterns.

**Correlation of environmental variables.** The Pearson correlation matrix of the environmental variables is shown in Table 8. Altitude was positively correlated with slope (r = 0.23*), disturbance (r = 0.35**), CEC (r = 0.36**), TN (**r = 0.26\*\***), and Av. P (r = 0.56*). Surprisingly, the slope was positively correlated with CEC (r = 0.02). Disturbances displayed negative correlations with SOC (r = -0.05*), TN (r = -0.25*), Av. P (r = -0.383**), CEC (r = -0.384**), and pH (r = -0.06) indicated that higher disturbance levels were associated with lower values of these variables. SOC exhibited a positive correlation with available phosphorus (r = 0.12),

**Table 7. The six axes show the biplot scores of the constraining variables, eigenvalues, and proportions of variance.**

| Variables | CCA1 | CCA2 | CCA3 | CCA 4 | CCA 5 | CCA 6 |
|---|---|---|---|---|---|---|
| Altitude | -0.95 | 0.11 | -0.24 | 0.03 | -0.05 | -0.09 |
| Disturbance | 0.30 | -0.94 | 0.00 | 0.01 | 0.14 | 0.05 |
| Slope | -0.11 | 0.17 | 0.02 | 0.47 | 0.23 | -0.12 |
| PH | 0.06 | 0.07 | 0.40 | -0.87 | 0.16 | -0.02 |
| CEC | -0.28 | 0.42 | -0.88 | -0.86 | 0.16 | -0.02 |
| TN | -0.47 | 0.21 | -0.27 | 0.03 | -0.03 | -0.82 |
| Av. P | -0.43 | 0.30 | 0.14 | 0.07 | 0.41 | 0.07 |
| SOC | -0.25 | 0.15 | -0.09 | -0.15 | -0.62 | 0.12 |
| Eigenvalue | 0.34 | 0.15 | 0.14 | 0.12 | 0.11 | 0.10 |
| Proportion explained | 0.30 | 0.13 | 0.12 | 0.11 | 0.10 | 0.09 |
| Cumulative proportion | 0.30 | 0.43 | 0.55 | 0.66 | 0.76 | 0.85 |

**Table 8. Pearson correlation coefficients between environmental variables in Arjo-Diga Forest, Eastern Ethiopia.**

|  | Altitude | Slope | Disturbance | SOC | TN | Av. P | pH | CEC |
|---|---|---|---|---|---|---|---|---|
| Altitude |  |  |  |  |  |  |  |  |
| Slope | **0.23*** |  |  |  |  |  |  |  |
| Disturbance | **0.35**** | -0.11 |  |  |  |  |  |  |
| SOC | **0.16*** | -0.23 | -0.05* |  |  |  |  |  |
| TN | **0.26**** | -0.15 | **-0.25*** | -0.03 |  |  |  |  |
| Av. P | **0.56*** | -0.1 | **-0.38**** | **0.12*** | -0.06 |  |  |  |
| pH | -0.45** | -0.19* | -0.06 | -0.07 | **-0.29**** | **-0.29**** |  |  |
| CEC | **0.36**** | 0.02 | **-0.38**** | **-0.01*** | -0.24* | 0.18 | **0.27*** |  |

*. Correlations are significant at the 0.05 level (2-tailed).

**. The correlation is significant at the 0.01 level (2-tailed).

implying that phosphorus availability tends to increase as the level of soil organic carbon increases. Conversely, it displayed negative correlations with CEC (r = -0.01*), TN (r = -0.17), and pH (r = -0.03). TN was negatively correlated with pH (r = -0.29**), CEC (r = -0.24*) and Av. P (r = .-0.06). Additionally, Av. P demonstrated a positive correlation with CEC (r = 0.01).

**Relationships between plant community types and environmental variables.** Based on the results obtained from the post hoc mean comparison, significant differences were found between the five plant community types concerning environmental variables (Fig 5 and Table 9). Community type 1 was found at a higher altitude (2011.8±0.3), while community type 5 was found at a lower (15285±0) altitude. Community type 5 exhibited the highest disturbance value (2±0.5), while community type 3 had the lowest (1.4±1.13). The highest CEC (40.3±6) was recorded for community type 2, while the lowest CEC (31.2±10.7) was recorded for community type 5. Community type 2 had the highest soil organic carbon value (4.19), while community type 5 had the lowest value (2.85). The pH values of the soils ranged from 4.8±0.5 to 5.1±0.4 among the different community types (Table 8). Consequently, community type 4 was found in highly acidic soils, while communities 1, 2, 3, and 5 were found in strongly acidic soils.

Additionally, significant differences were observed in species richness, diversity, and evenness along environmental gradients, as depicted in Fig 6. The results indicated that both species richness ($R^2$ = 0.07, p < 0.05) (Fig 6A) and Shannon diversity ($R^2$ = 0.05, p < 0.05) (Fig 6B) decreased significantly with increasing altitude. Species richness displayed a negative association with TN ($R^2$ = 0.04, p<0.05), suggesting that more significant TN deposition led to a decrease in species richness (Fig 6E). Conversely, the slope exhibited a significant negative

**Table 9. Post hoc comparison of means between environmental variables and plant communities.**

| Environmental variable | Plant community Types | | | | |
|---|---|---|---|---|---|
|  | C1 | C2 | C3 | C4 | C5 |
| Altitude(m) | 2011.8±0.3[d] | 1761.4±1.2[a] | 1917.4±2.1[c] | 1648.1±.2.5[b] | 1528.5±0.5[b] |
| Slope (%) | 21.8±.3.6[bc] | 19.8±11[bc] | 22.7±1.3[c] | 21.1±5[a] | 20.3±1[a] |
| Disturbance | 1.85±1[ab] | 2±0.6[ab] | 1.4±1.13[a] | 1.57±0.7[a] | 2±0.5[ab] |
| pH | 5.1±0.4[a] | 5.08±0.7[a] | 5.02±0.5[a] | 4.8±0.5[b] | 5.1±0.3[a] |
| CEC | 38.8±7.3[c] | 40.3±6[b] | 39.8±8[b] | 32.9[a] | 31.2±10.7[a] |
| SOC | 3.6[b] | 4.19[a] | 3.19[b] | 4.08[a] | 2.85[c] |

Note: Values in a row with different letters differ significantly (P<0.05).

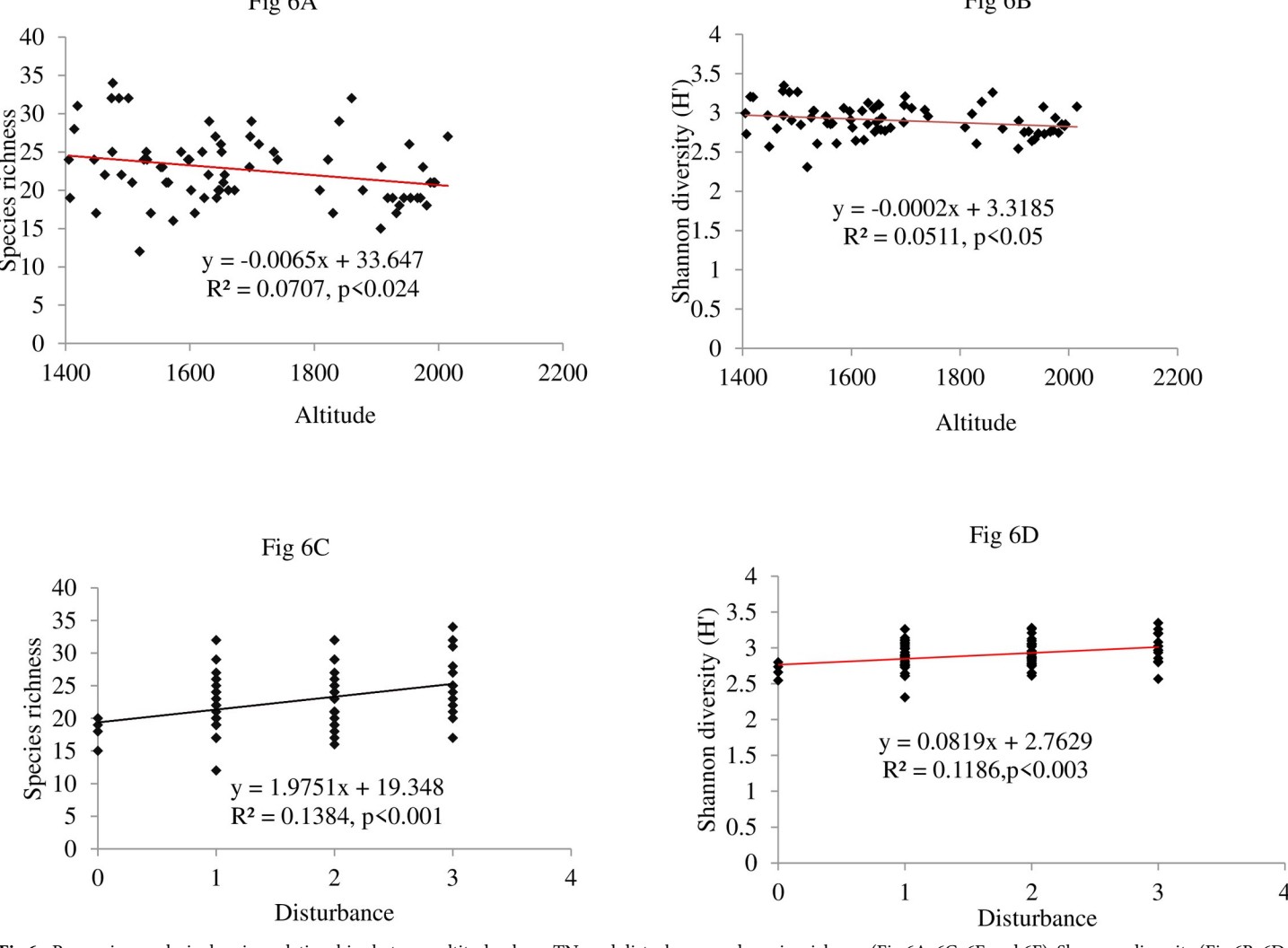

**Fig 6.** Regression analysis showing relationships between altitude, slope, TN, and disturbance and species richness (Fig 6A, 6C, 6E and 6F), Shannon diversity (Fig 6B, 6D and 6G), and evenness (Fig 6H).

association with Shannon diversity ($R^2 = 0.01$, $p < 0.01$) (**Fig 6G**) and evenness ($R^2 = 0.08$, $p < 0.01$) (**Fig 6H**), indicating that steeper slopes were linked to lower diversity and evenness. Interestingly, the disturbance had a significant positive association with species richness ($R^2 = 0.11$, $p < 0.003$) (**Fig 6C**) and Shannon diversity ($R^2 = 0.13$, $p < 0.001$) (**Fig 6D**), suggesting that areas with higher disturbance levels exhibited greater species richness and species diversity.

## Discussion

We observed relatively greater species richness (234 vascular plant species) in the Arjo-Diga forest than in other similar studies conducted in Ethiopia, e.g., in Masha forest (130 species [44]; Jibat forest, 183 species [45]; Komto forest, 180 species [46]; Belete forest, 157 species [47]; Gelesha forest, 157 species [48]; Gendo forest, 168 species [49]; Agama forest, 162 species [50]; and Gerba Dima forest, 180 species [23]. However, the species richness of the study forest was lower than that reported by Kelbessa and Soromessa [51] in the Bonga forest (243 species).

The variation in species composition observed between different forests can be attributed to the number of plots sampled, and their size may, to some extent, illustrate the heterogeneity of species richness. A study by Yirga et al. [52] revealed that forests subjected to significant human interference and disturbance tends to have lower species richness than others. Thus, the present study showed that the Arjo-Diga forest has a greater species composition than similar vegetation types in Ethiopia. Asteraceae and Fabaceae, Rubiaceae, Euphorbiaceae, Lamiaceae, Acanthaceae, and Poaceae are the top ten most species-rich families in many Neotropical forests [53], flora areas [54] and other moist Afromontane forests of southwestern Ethiopia [23,47,52]. Thus, the dominance of these families in the Arjo-Diga Forest agreed with their general dominance in the flora area and tropical forests. The dominance of these families could be attributed to their adaptation and colonization of different ecological niches based on their efficient pollination, dispersal, and germination mechanisms [55]. For instance, Fabaceae species possess various advantageous traits, such as seed resistance to predators, the ability to recover leaves and branches after defoliation, and the ability to germinate rapidly in the presence of moisture [56]. The high herbaceous species recorded are more likely to be explained by the open nature of the vegetation canopy, which allowed ground-level plants to grow freely. This observation aligned with the findings reported by Murphy and Lugo [57], who noted that the abundance of herbaceous species tended to be inversely related to the degree of canopy cover in the vegetation. The dominance of shrubby species in moist Afromontanae forests is due to the selective cutting of trees [47,58], and other anthropogenic factors cause the dominance of herbaceous species in a specific area.

Approximately 8.54% of the plant species are endemic to Ethiopia and Eritrea. This is lower than the previously reported 10–15% endemism in dry Afromontane forests [59]. The lower proportion of endemic plant species observed in the forest is attributed to the significant disturbances from human activities and livestock grazing in the study area. Ethiopia's moist Afromontane forest vegetation faces considerable environmental stress [29,60]. Consistent with this finding, other studies [61] have also reported a low occurrence of endemic plant species in moist Afromontane forests. Adopting an ecosystem-based approach to biodiversity conservation and participatory forest management holds tremendous potential for safeguarding numerous rare and endemic species [62,63].

## Plant community types

Hierarchical cluster analysis identified five distinct plant communities in the study area. However, communities 2, 3, and 4 share overlapping elevation ranges. Elevation is a complex gradient encompassing various environmental factors, including topography, climate, and soil variables [64]. Consequently, isolating other environmental factors becomes challenging due to the interrelated nature of these variables.

The study revealed that each identified plant community had a different floristic composition, and this variation could be due to differences in environmental factors. A survey by Adal [65] reported that differences in species composition among plant communities may be associated with environmental factors. However, it is essential to note that species diversity and richness were not uniform across plant communities. There were inherent differences in these measurements between the different plant communities, suggesting unique ecological characteristics and conditions within each community. For example, the study revealed that communities 2 and 4 had high species richness and diversity. In contrast, community 3 had the lowest species richness and diversity, likely due to the influence of various human activities in that area, including livestock grazing, charcoal production, proximity to human settlements, and firewood collection [66]. The possible reason for community three's high species diversity and

richness may be its location within the middle altitudinal range from 1500 to 1700 m.a.s.l. This intermediate altitudinal habitat appears to provide favorable conditions that enable the rapid acquisition of resources and support the flourishing growth of a diverse array of plant species within this community.

Anthropogenic factors are major drivers of global biodiversity change [67]. Some plant species may experience a steady or more rapid deterioration in their environmental conditions due to changes in land-use practices, which could decrease their abundance and distribution. For example, over the last 20 years, the forest coverage in the Diga district has decreased due to farming, grazing, and settlement [26]. Therefore, the alterations in anthropogenic land use are likely to influence the composition of the plant community in the present study, resulting in changes in both plant richness and diversity.

The analysis of Sorensen's similarity coefficient index revealed a significant level of similarity in species composition among three plant communities, Community 1, Community 2, and Community 4 because they share similar locations and environmental factors, such as soil characteristics and topography [68]. Community 5 is located at a lower altitude (1470 m.a.s.l.) with low organic matter content, soil nutrients, and moisture content, which may result in lower floristic similarity than other plant community types (communities 1, 2, and 4).

## Environmental variables and plant community relationships

Understanding the relationships between plant community composition and environmental variables is crucial for understanding community patterns in forest ecosystems [69,70]. The species composition of plant communities is influenced by several environmental factors, such as soil, geography, climate, and human disturbances [62,71]. Several ecological studies conducted in Ethiopia have reported that environmental variables are essential for shaping plant communities [38,62,72,73].

Canonical correspondence analysis (CCA) result analysis revealed that the plant communities in the study forest were strongly influenced by topographic factors such as altitude and slope, as well as edaphic variables such as pH, cation exchange capacity (CEC), and total nitrogen (TN). Additionally, anthropogenic disturbances were found to impact plant communities significantly. Specifically, CCA axis 1 was correlated with disturbances, while CCA axis 2 was correlated with altitude, CEC, pH, and TN. The second axis (CCA) accounted for approximately 13% of the total variance, while the first axis (CCA) explained 30% of the variance. CCA1 and CCA2 accounted for approximately 43% of the total variance (inertia). These findings indicate that elevation, CEC, TN, soil pH, and disturbance play crucial roles in influencing the distribution of plant species and the formation of plant communities in the study area. These results align with previous research emphasizing the significant impact of topographic and soil variables on plant species distribution and community formation [72,74,75].

Elevation plays a crucial role in accounting for differences in the distribution of plant species and the formation of plant communities, with some communities showing overlapping characteristics. This can be attributed to the gradual changes in environmental variables that occur along the gradient of elevation [76]. Similar studies conducted by researchers in various regions of Ethiopia have also found elevation to be an important topographic variable in determining patterns of vegetation distribution.

Edaphic factors (soil variables), such as TN, soil pH, and CEC, also significantly affected the growth and distribution of the plant communities in this study. Changes in soil parameters exert a substantial influence on the development of plant communities [77]. Additionally, the chemical and physical attributes of the soil are interconnected with its inherent characteristics, thereby affecting the composition of plant species and the distribution of higher vascular

plants [78]. For instance, TN explained a significant variation in species composition within communities 1 and 3 (Fig 4) due to its positive effects on certain highly competitive plants, leading to the exclusion of species in competition. The findings of this study align closely with the results of a previous study conducted by Rawal [79], which also reported a negative relationship between TN levels and species diversity in plant communities within Pennsylvania forests. Similarly, a study by Shen et al. [80] reported that TN deposition influences plant community composition in European acidic grassland ecosystems. CEC was also a significant factor in explaining the differences in species composition between communities 1 and 2 in the current study. Similar to our results Zheng et al. [81] reported that CEC affects tree species in a plant community by affecting soil fertility. In addition, Bekele [72] found that CEC had a positive and significant impact on the species composition of the plant community on the central plateau of Shewa, Ethiopia. The results of this study suggest that the structure of plant communities is significantly influenced by soil chemical and physical properties.

Soil pH is a crucial factor that shapes the soil's biogeochemical processes, which directly affects the composition and diversity of plant communities and their productivity [82]. This study found plant community type 4 in soil with a pH of 4.8. At this low pH, the availability of essential macronutrients, such as nitrogen, phosphorus, and potassium, is reduced, leading to nutrient deficiencies and stunted plant growth and development within the community. Several studies have also shown that soil pH can influence nutrient availability, ultimately affecting the uptake of vital nutrients for plant growth and development [83–85]. However, it is essential to note that the availability of specific nutrients due to pH can adversely affect particular plants, as some nutrients can be toxic to them [86,87].

In addition to topographic and edaphic variables, anthropogenic disturbances, including tree logging, cattle grazing, and firewood collection, were identified as significant factors influencing species distribution patterns and community formation in the current study. The observed effects can be attributed to alterations in species richness, diversity, distribution patterns, and vegetation structure within the ecosystem due to disturbances [88,89]. The present study revealed that community 5 experienced the highest level of disturbance, primarily due to its proximity to neighboring agricultural activities and human settlements. Consequently, this disturbance led to a reduction in species richness within the plant communities. Additionally, a study revealed the presence of two local spice species (*Piper capense* and *Aframomum corrorima*), which are wild spices, in the sampled plots, indicating the presence of human intervention. Similarly, a study conducted by Tabarelli et al. [90] in tropical forests revealed that disturbances negatively impact seed dispersal and seedling formation, leading to changes in the distribution patterns of plant communities in these ecosystems.

Based on the simple regression analysis, we discovered a significant relationship between environmental variables and both species richness and species diversity. Specifically, altitude was significantly associated with species richness and diversity in this study. Altitudinal gradients are typically linked to variations in precipitation and temperature [91,92]. The observed pattern of species richness, characterized by a peak followed by a decrease, suggested that it may be influenced by the combined effects of local physical factors, such as soil properties, along with nonrandom fluctuations in temperature and precipitation along the elevation gradient.

Slope also plays a significant role in determining plant distribution due to its impact on the accumulation and export of soil nutrients. Previous research conducted by Zhang et al. [93] has shown that plant diversity is influenced by slope, as the lower part of the slope provides a favorable habitat with rich soil and water conditions, resulting in high plant diversity. Conversely, steeper slopes exhibit poor soil and water conditions, leading to low plant diversity. Other studies, such as [74,94], have demonstrated that slope gradients significantly impact

species diversity and evenness by affecting crucial ecological factors such as humidity, heat, light intensity, and soil conditions. Several studies have also reported that a greater slope inclination exacerbates conditions that prevent competitive species from monopolizing resources [95–97]. This phenomenon increases plant diversity, as broader species can coexist in challenging environments.

### Correlations among environmental variables

In the Arjo-Diga forest, SOC was positively correlated with altitude, suggesting that SOC values increase with increasing altitude. The influence of altitude on plant communities is complex and likely indirect. Typically, temperature decreased with increasing altitude, while precipitation showed an upward trend. These climatic changes along elevation gradients affect vegetation composition and productivity and subsequently affect the amount and turnover of SOC [98]. The decrease in temperature associated with higher altitudes is likely to result in lower SOC turnover rates and possibly an increase in SOC levels. This temperature-related effect can influence the accumulation and stability of soil organic carbon and contribute to higher SOC at higher altitudes [99]. A study in the Gerba-Dima forest by Dibaba et al. [100] also supported the positive correlation between altitude and SOC. However, it is important to note that a study by Shapkota and Kafle [101] reported contradictory results, as they reported a decrease in SOC with increasing altitude. This discrepancy could be due to reduced decomposition rates associated with organic carbon production and long-term accumulation. Despite this discrepancy, the current study confirms that elevation is a reliable indicator of forest soil organic carbon content.

Moreover, a positive correlation was observed between the cation exchange capacity (CEC) and altitude. This association could be explained by the higher vegetation density at higher altitudes, which leads to increased organic matter production and, consequently, higher CEC [102]. Similarly, a study on tea plantations in Indonesia reported that adding organic matter to the soil can improve soil CEC. However, Zhang et al. [78] have reported a negative correlation between CEC and SOC, as decreased soil organic matter (SOM) decomposability leads to decreased CEC. SOC and TN exhibit a positive correlation due to their common origin in organic matter. Nitrogen is released as SOM decomposes, increasing the TN content in soils with high SOC. Furthermore, there is a negative correlation between soil pH and SOC, as soil pH plays a crucial role in regulating the diversity of microbial communities that affect the rate of SOC degradation. By understanding the interrelationships among these soil properties, appropriate soil management practices can be implemented to enhance soil fertility and support diverse plant communities.

### Conclusion

This study investigated the floristic composition, species diversity, and relationships between plant communities and environmental variables in the Arjo-Diga forest. Five distinct plant community types were identified via agglomerative hierarchical cluster analysis, each exhibiting different diversity index values. Community type 5 demonstrated the highest species richness and diversity among the identified types. Altitude was recognized as the most significant factor influencing species composition, diversity, and community formation, followed by cation exchange capacity (CEC), total nitrogen (TN), and disturbance. These factors played crucial roles in shaping the characteristics of the plant communities and their associated diversity. Considering the combined effects of various environmental factors is vital for a comprehensive understanding of the variations in species richness, diversity, and evenness within plant community types and the distribution of plant species in a specific area. This study also highlighted

the substantial impact of anthropogenic disturbances and the strong dependence of the local community on forest resources. Consequently, conservation efforts should prioritize these areas to safeguard ecologically important species, including the twenty endemic taxa identified. These findings provide valuable insights for guiding conservation and management practices in the Arjo-Diga Forest.

## Supporting information

**S1 Table. Floristic list of species in Arjo-Diga forest.**
(DOCX)

**S2 Table. Value of the indicator species in identified plant communities and their significant p value.**
(DOCX)

## Acknowledgments

The authors would like to express their deepest gratitude to the Diga District Agriculture and Rural Development Department and the local community surrounding the forest for their invaluable support in field data collection. In addition, the authors would like to thank the anonymous reviewers of PLOS ONE for their valuable comments and feedback.

## Author Contributions

**Conceptualization:** Zemede Asfaw Woldemariam.

**Supervision:** Tamrat Bekele Gode, Ermias Lulekal Molla, Zemede Asfaw Woldemariam.

**Writing – original draft:** Tariku Berihun Tenaw.

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
