## [Decision Letter · Decision Letter 0]

18 May 2023

PONE-D-23-07720Topography factors and soil variables drive the plant community distribution pattern and species richness in the Arjo Diga forest in western Ethiopia.PLOS ONE

Dear Dr. berihun,

Thank you for submitting your manuscript to PLOS ONE. After careful consideration, we feel that it has merit but does not fully meet PLOS ONE’s publication criteria as it currently stands. Therefore, we invite you to submit a revised version of the manuscript that addresses the points raised during the review process. After the editorial evaluation, the MS was sent to two expert reviewers. Both the reviewers have suggested significant revision in terms of technicality, data analysis, presentation style, and English language. Please look into the comments very carefully and revised the MS thoroughly. Major part of the MS needs rewriting, and through correction of English is a must. The revised version will be sent for external reviews once again, before making the decision on the MS.

Please submit your revised manuscript by Jul 02 2023 11:59PM. If you need more time than this to complete your revisions, please reply to this message or contact the journal office at plosone@plos.org. Please include the following items when submitting your revised manuscript:A rebuttal letter that responds to each point raised by the academic editor and reviewer(s). You should upload this letter as a separate file labeled 'Response to Reviewers'.A marked-up copy of your manuscript that highlights changes made to the original version. You should upload this as a separate file labeled 'Revised Manuscript with Track Changes'.An unmarked version of your revised paper without tracked changes. You should upload this as a separate file labeled 'Manuscript'.

We look forward to receiving your revised manuscript.

Kind regards,

Bhoj Kumar Acharya, PhD

Academic Editor

PLOS ONE

“The authors thank Dilla University for giving the first author the opportunity to continue his doctoral studies. Funding from Dilla University and Addis Ababa University is also recognized. We would also like to thank the Diga District Department of Agriculture for providing the necessary information for this study. We also thank all the staff of the Ethiopian National Herbarium for their assistance in identifying the flora. We would also like to express our gratitude to the local authorities and community for their kind support and cooperation throughout the data collection”

6. We note that you have referenced (Hadera, G., A study on the ecology and management of the Dessa forest in the northeastern escarpment of Ethiopia. Unpublished MSc Thesis Addis Ababa University, 2000.) which has currently not yet been accepted for publication. Please remove this from your References and amend this to state in the body of your manuscript: (ie “Bewick et al. [Unpublished]”) as detailed online in our guide for authors

7. Please include a separate caption for each figure in your manuscript.

8. We note that Figure 1 in your submission contain [map/satellite] images which may be copyrighted. All PLOS content is published under the Creative Commons Attribution License (CC BY 4.0), which means that the manuscript, images, and Supporting Information files will be freely available online, and any third party is permitted to access, download, copy, distribute, and use these materials in any way, even commercially, with proper attribution. For these reasons, we cannot publish previously copyrighted maps or satellite images created using proprietary data, such as Google software (Google Maps, Street View, and Earth). For more information, see our copyright guidelines: http://journals.plos.org/plosone/s/licenses-and-copyright.

Reviewers' comments:

Reviewer's Responses to Questions

**Comments to the Author**

1. Is the manuscript technically sound, and do the data support the conclusions?

Reviewer #1: No

Reviewer #2: Yes

2. Has the statistical analysis been performed appropriately and rigorously? 

Reviewer #1: Yes

Reviewer #2: Yes

3. Have the authors made all data underlying the findings in their manuscript fully available?

Reviewer #1: Yes

Reviewer #2: Yes

4. Is the manuscript presented in an intelligible fashion and written in standard English?

Reviewer #1: No

Reviewer #2: Yes

5. Review Comments to the Author

Reviewer #1: The work conducted by the authors is appreciated. However, the current draft of the manuscript is lacking the standards of the journal and it looks prepared in hurried manner. Number of typos are clearly visible throughout the submitted manuscript daft, a few of them are marked in the attached manuscript draft. Most of figures does not possess figure legends. The manuscript needs a major revision in terms of its presentation, prior to its consideration for the publication by the journal.

Thank you

Reviewer #2: The MS is interesting and readable but it needs to be improved. I have provided report in the attachment. I suggest do some regression analysis with elevation data and draw the pattern.

Topography factors and soil variables drive the plant community distribution pattern and species richness in the Arjo Diga forest in western Ethiopia.

The present MS is about a unique type of study made by four Ethiopian authors. They focus on community analysis along 106 environmental gradients and seek to assess the impact of environmental factors on plant diversity (i.e., richness, diversity and evenness). The MS seems interesting and worth-full to read as well as I enjoyed reading it. The data have been collected by authors by sampling which add more value on the MS. This sort data base may add credit to define species diversity and composition theory in the field of ecology.

Beside this I found some weak points. The major weakness of the study is not focused well as per the aims of the study. When one going for studies the MS he or she may confuse whether it is local scale study or macro scale study. The Arjo-Diga forests have elevations gradients ranging from 1,200 to 2,220 meters above sea level and I ask the authors to show the patterns along the elevation gradient as well?

However, there are some more flaws in the MS which needs to be improved before publication of the MS. I have jotted down some of the flaws in the MS as below:

Abstract:

The abstract is relatively longer and it confusing so I suggest to rephrase it.

Introduction:

The study has two types of data set like elevation gradient which is macro scale factor and soil, disturbance etc are local scale factors so the authors have to discuss about these factors how they influence the species patterns. So they have to demonstrate how these factors are associated in their study.

Methodology

The Arjo-Diga has elevations ranging from 1,200 to 2,220 meters above sea level which quite long gradient so I suggest honoring this gradient. Why authors use DCA and CCA have to elaborate why this data set suitable to use these techniques. They have not provided DCA and CCA diagram and their interpretation.

Others part of methodology is okey.

Result:

The authors have identified 5 types of plant communities I still want to see these communities in the ordination diagram. So I want to ask to make CCA diagram how these five types of communities distributed?

Discussion

I am skeptical with the presentation of discussion. The subtopics which are provided are vague so I want to see discussion under the specific subheading like: relationship between species diversity and climatic variables or relationship between species composition and environmental factors etc…. and how their finding is similar or difference with others such studies in the different scales in the Ehiopia and other parts of Globe.

Conclusions

The conclusion rephrased as it is bit confusing!

In conclusion I found the MS is interesting but it still needs to be revised thoroughly.

6. PLOS authors have the option to publish the peer review history of their article (what does this mean?). If published, this will include your full peer review and any attached files.

Reviewer #1: No

Reviewer #2: **Yes: **Khem Raj Bhattarai

---

## [Author Response · Author response to Decision Letter 0]

30 Jul 2023

we adressed all comment and suggestion by editor and two reviwers and upload it separately

---

## [Decision Letter · Decision Letter 1]

14 Sep 2023

PONE-D-23-07720R1Topography factors and soil variables drive the plant community distribution pattern and species richness in the Arjo Diga forest in western Ethiopia.PLOS ONE

Dear Dr. Berihun,

Thank you for submitting your manuscript to PLOS ONE. After careful consideration, we feel that it has merit but does not fully meet PLOS ONE’s publication criteria as it currently stands. Therefore, we invite you to submit a revised version of the manuscript that addresses the points raised during the review process. The revised version was reviewed by three experts (earlier two reviewers and one additional reviewer). All of them have positively commented on the MS but  there are few important aspects that need further improvement. Reviewer 3 have given some useful comments for improvement, and hence, authors are requested to critically look into the comments and revise the MS carefully. Please also look for typos, grammar, formats (including maps, figures and tables) as well as uniformity in presentation. 

We look forward to receiving your revised manuscript.

Kind regards,

Bhoj Kumar Acharya, PhD

Academic Editor

PLOS ONE

Reviewers' comments:

Reviewer's Responses to Questions

**Comments to the Author**

1. If the authors have adequately addressed your comments raised in a previous round of review and you feel that this manuscript is now acceptable for publication, you may indicate that here to bypass the “Comments to the Author” section, enter your conflict of interest statement in the “Confidential to Editor” section, and submit your "Accept" recommendation.

Reviewer #1: All comments have been addressed

Reviewer #2: All comments have been addressed

Reviewer #3: All comments have been addressed

2. Is the manuscript technically sound, and do the data support the conclusions?

Reviewer #1: Yes

Reviewer #2: Yes

Reviewer #3: Yes

3. Has the statistical analysis been performed appropriately and rigorously? 

Reviewer #1: Yes

Reviewer #2: Yes

Reviewer #3: Yes

4. Have the authors made all data underlying the findings in their manuscript fully available?

Reviewer #1: Yes

Reviewer #2: Yes

Reviewer #3: Yes

5. Is the manuscript presented in an intelligible fashion and written in standard English?

Reviewer #1: Yes

Reviewer #2: Yes

Reviewer #3: Yes

6. Review Comments to the Author

Reviewer #1: The revised manuscript looks in good shape and authors have adequately addressed endorsements in the manuscript.

A small correction needs to be incorporated in the manuscript.

Instead to A study by [7] found different patterns of species richness along different elevation gradients.

authors should write A study by Pandey et al [7] found different patterns of species richness along different elevation gradients.

Rest is Ok.

With Best Regards

Reviewer #2: The MS has been improved but I am still skeptical with MAP showing the location of Ethiopia in the Africa.

Reviewer #3: The study titled "Topography factors and soil variables drive the plant community distribution pattern and species richness in the Arjo Diga forest in western Ethiopia" aims to examine three critical points.

(1) the floristic composition and diversity of the Arjo Diga forest,

(2) the influence of environmental variables on the characteristics of plant diversity

(Richness, variety, and uniformity)

(3) Investigate the relationship between plant community types and environmental variables.

Such a kind of study, especially from the underrepresented areas, is essential and requires our attention. This manuscript has valuable finding which is vital for protecting existing plant species in the Arjo-Diga forest in Western Ethiopia.

The authors have put a reasonable effort into addressing reviewers' comments; however, there is a scope to make more corrections, which can be quickly addressed and will improve the manuscript's readability. The discussion sections require some improvements.

While revising, please be careful of typos; I have highlighted some of them. Also, edit the tables and figures following a similar format throughout the manuscript.

Here are my detailed comments.

Abstract:

L17-18, The plant family information is in between sentences, which discusses plant communities. The author can start with plant families, followed by another sentence about plant communities or vice versa.

Introduction:

The full form of TN and CEC needs to be included in the introduction.

L72-77 It would be nice if author could provide references for the following sentences

Materials and Methods

L113 Here, the author may introduce meters above sea level abbreviation as (m.a.s.l.), which can be used for the rest of the sections.

Throughout this section, authors can cite the references directly instead of using according to ……

Ethical statement

Although it is an important point, I am unsure if the ethical statement must be a part of the methodology section.

If it is only about the permits/permission the authors required for the sample collection, this will fit well in the acknowledgements.

Methods of data collection

L162: Use abbreviations (m.a.s.l.)

L165: Does it mean that out of 72 plots, a single plot size was 30 m x 30 m (900 m2) and inside each such plots authors also laid 5 m x 5 m (25 m2) and five 2 m x 2 m (4 m2) to record other life forms such as shrubs and herbs? If it is so, then it needs to be rewritten. Also, unlike herbs, how many 5 m x 5 m (25 m2) were laid in each plot needs to be clearly mentioned.

L169: Does it mean that voucher specimens per species were collected from all plots for unidentified/identified species?

Environmental Data

L179: Won’t the soil samples be more than 72? If it is collected from five different locations at each 1 m x 1 m subplot. Please clarify.

L181 airdried in natural conditions? Or a hot air oven?

L184 The full form and abbreviations of TN and CEC should be mentioned in the introduction.

L191-194: It will be easier for a reader to understand the methodology if we give more information than just citing previous papers. The scale looks interesting, but it will be more apparent if the authors briefly describe the signs of disturbance that were considered for data collection. For example, the presence or absence of cattle dung? Proximity to the forests?

Also, reading the discussion, human-caused disturbance is one of the major causes of species loss, so adding more information to this section would be worthwhile.

Results

Table 2: Having the full form of the floristics regions in the caption will be good.

Also, were any of these species endemic? Will it be possible to add a row showing whether these are endemic? Ignore this comment if endemic species are listed elsewhere in another table.

L266: Package information and its reference must be mentioned in the methods/analysis section, not the results section.

L370: Delete however and use while to connect both the sentences

L373 attitude?

L375 axes are 0.3504 and 0.2346 ….. follow the similar no of digits after

the decimal points for the entire manuscript.

Table 7 Altitude (m a s l) needs to be replaced with m.a.s.l.

Also, the row title is in bold, while the rest of the tables are not; please follow the same formatting style.

The current file shows all the figures as Fig 1, which needs to be clarified. Is it the formatting issue?

Kindly have a look at it and make the required changes.

Fig - Dendrogram of vegetation data from agglomerative hierarchical cluster analysis-

In this figure, the legend and the caption for the y and x axes need a larger font size; it is now difficult to read.

Fig- Regression analysis along altitude, slope, TN, and disturbance.. Species richness ----- For this figure, the significance can be mentioned

at a similar location for all figures, towards the bottom right or top right. That will make the figure look clean and easy to read. Furthermore, please follow similar width and breadth sizes for all the figures.

Fig 1. Influence of environmental variables on plant community types- here, please mention the plant community types.

Discussion

The first paragraph of the discussion should help the reader recollect your aims or the questions addressed, followed by the data used and the results found. Overall, the first paragraph of the discussion sections will lead to the remaining section of the discussion. I wonder if the comparison with the other forests at the beginning of the discussion makes any sense. It distracts the readers from the main questions.

Either remove it or reshuffle the flow of this section.

7. PLOS authors have the option to publish the peer review history of their article (what does this mean?). If published, this will include your full peer review and any attached files.

Reviewer #1: **Yes: **Dr. Aseesh Pandey

Reviewer #2: **Yes: **Khem Raj Bhattarai

Reviewer #3: **Yes: **SHWETA BASNETT

---

## [Author Response · Author response to Decision Letter 1]

18 Oct 2023

We thank all three reviewers who gave us very constructive comments to better improve our manuscript. Based on the reviewers' comments and suggestions, we consider all the three reviewers' comments and suggestions and attach them to the three reviewers individually.

---

## [Editor Report · Decision Letter 2]

25 Oct 2023

PONE-D-23-07720R2Topography factors and soil variables drive the plant community distribution pattern and species richness in the Arjo Diga forest in western Ethiopia.PLOS ONE

Dear Dr. Berihun,

Thank you for submitting your manuscript to PLOS ONE. After careful consideration, we feel that it has merit but does not fully meet PLOS ONE’s publication criteria as it currently stands. Therefore, we invite you to submit a revised version of the manuscript that addresses the points raised during the review process. The MS is significantly improved compared to previous version but still there are issues that needs to be addressed. Please look into the comments of Reviewer 2 on Map of the study area and all the comments of Reviewer 3. Revise the MS carefully and critically based on the comments. Please look into English Grammar, Typos and formatting of text as well as tables and figures. 

We look forward to receiving your revised manuscript.

Kind regards,

Bhoj Kumar Acharya, PhD

Academic Editor

PLOS ONE

---

## [Author Response · Author response to Decision Letter 2]

4 Dec 2023

We have submitted the revised manuscript as per the comments provided by the editors,and reviewers . Thank you for your guidance and feedback throughout the review process. We appreciate the opportunity to improve our manuscript and address the concerns raised

---

## [Editor Report · Decision Letter 3]

18 Dec 2023

PONE-D-23-07720R3Topography factors and soil variables drive the plant community distribution pattern and species richness in the Arjo Diga forest in western Ethiopia.PLOS ONE

Dear Dr. Berihun,

Thank you for submitting your manuscript to PLOS ONE. After careful consideration, we feel that it has merit but does not fully meet PLOS ONE’s publication criteria as it currently stands. Therefore, we invite you to submit a revised version of the manuscript that addresses the points raised during the review process. The MS is improved compared to previous version but still there are series of issues. Please look into comments from the editor in the attached MS. Please look into the comments thoroughly and revise the MS. Although the MS has undergone several rounds of review but many of the comments are still not addressed. Hence, this is the last chance for revision, and if the MS is not carefully revised there may be a possibility of rejection without further review.

We look forward to receiving your revised manuscript.

Kind regards,

Bhoj Kumar Acharya, PhD

Academic Editor

PLOS ONE

Additional Editor Comments: 

The MS is improved compared to previous version but still there are series of issues. Please look into comments in the attached MS. Please look into the comments thoroughly and revise the MS. Although the MS has undergone several rounds of review but many of the comments are still not addressed. Hence, this is the last chance for revision, and if the MS is not carefully revised there may be a possibility of rejection without further review.

---

## [Author Response · Author response to Decision Letter 3]

16 Feb 2024

Dear Editor,

We sincerely appreciate the valuable feedback provided by the editors. Taking their comments into careful consideration, we have made significant improvements to the manuscript. The revised version incorporates the suggested changes and addresses the editors' concerns. We are immensely grateful to the editors for their insightful comments and suggestions, as they have undoubtedly enhanced the quality of this study.

In response to the specific comments raised by the editors, we have provided a detailed point-by-point response below

Comment from editors 

Introduction 

Comment 1: Line 114-116.what is the difference between objective 1 and 2?, I think these question is objective 2 and 3 because the highlight yellow color is objectives 2 and 3. Therefore we respond objectives highlighted by yellow color objectives 2 and 3. 

Response 1. Thank you for pointing this out. The two objectives are similar, so we have merged them based on the editor's comment.

Materials and methods

Comment 2: Line 129 pleases check. This is not the way of writing. There are so many such issues in the MS, and the problem was pointed by the reviewers but still not solved.

Response 2: We apologize for the mistakes we made. We have now made the necessary corrections based on the editor's comment.

Comment 3: Line 147 figure legends should be more elaborative which should stand alone

Response 3: Thank you. Based on the comment, we have elaborated the figure legends to make them more comprehensive and able to stand alone.

Comment 4: Line 206-2016 this should be a part of data analysis but not the data collection. Please rectify 

Response 4: Thank you for pointing that out. Based on the comment, we have moved the paragraph to the data analysis section.

Data analysis

Comment 5: line 239 Where is S in the formula? Please check. Moreover, all these formulae are universally known and no need to provide in the MS

Response 5: The letter 'S' in the summation formula represents a variable. To improve clarity, we were provided a list of abbreviations and their corresponding meanings. This would enhance understanding of the variables and their significance within the study..

Comment 6: Line 252-255 It is already known to the scientists

Response 6: Thank you for your feedback. We have acknowledged and modified the sentence accordingly.

Comment 7: Line 259-260 This is also not a part of ordination??? 

Response 7: Thank you for your feedback. We have revised the sentence to remove it from the section on ordination and present it as an independent statement, improving readability of the text

Result 

Comment 8: Line 276 the titles should be elaborative in nature which should stand alone. Please check all the tables carefully

Response 8: Thank you for your feedback. We agree that the table captions should be more comprehensive and able to stand alone. We have also provided the full meaning of table abbreviations

Comment 9: Line 293- 295 Please see my comments for title details

Response 9: Thank you for your feedback. We have carefully reviewed the titles of the tables and made the necessary corrections based on your comments..

Comment 10: line 297-298 This table can be shifted as supplementary material

Response 10: Thank you for your suggestion. We agree and have moved the tables to the supplementary material.

Comment 11.line 363-381 This section looks so weak and does not add any value to the MS. Combine this section and similarity and make a new section as community parameters and write in a more advance way. 

Response 11: Thank you for your feedback. As suggested, we have combined and rewritten these sections to create a new section on community parameters, which provides a more advanced and valuable analysis.

Comment 12. Line 386 please check 

Response 12. We have reviewed and rewritten the whole paragraph to enhance its quality.

Comment 13: Thank you for your feedback. We have carefully reviewed the sections and agree that they are similar. We will combine them accordingly.

Response 13: Thank you pointing this out. As you said it is almost similar with the above section and agreed combine them 

Comment 14..Line 466, I think this analysis is just a repetition of the results presented in earlier section. Regression and correlation analysis both are not needed for the same datasets

Response 14: We appreciate the editor's feedback. After careful consideration, we agree that the analysis in this section may overlap with the earlier results. To avoid redundancy, we will exclude the regression and correlation analysis for the same datasets. Thank you for bringing this to our attention, and we will revise the section accordingly.

Discussion

Comment 15: line 486-487, Discussion will change after the modification of the results. Hence, revise the discussion accordingly

Response 15: We understand that modifying the results will impact the discussion section. We have revised the discussion to reflect the changes made in the results section. By excluding redundant ideas and sentences, we have focused the discussion on the most relevant findings and their implications. Thank you for your guidance, and we have made the necessary adjustments to the discussion section.

Comment 16: line 512 ?????

Response 16: Thank you for pointing that out. We have corrected the miswrittenletter "t" before reference [64].

Comment 17: Line 512 this finding is consistent with the findings of Please see. This is not the correct way of writing scientific text. Many such problems occur in the text. Please see thoroughly and rectify

Response 17: Thank you for bringing this to our attention. We apologize for the incorrect usage in the scientific text. We will thoroughly review the entire document and rectify any similar issues. As per your suggestion, we will revise the text appropriately to maintain scientific rigor and clarity. We appreciate your guidance, and we will make the necessary revisions accordingly.

Comment 18: ??

Response 18: Thank you for your feedback. We apologize for the incorrect usage in the text. We were revised the sentence.

Comment 19: I do not see any significant addition of knowledge from this section. Please rewrite

Response 19: Thank you for your feedback. We will carefully revise this section to ensure that it adds significant knowledge and contributes to the overall findings of the study.

---

## [Decision Letter · Decision Letter 4]

28 May 2024

PONE-D-23-07720R4Topography factors and soil variables drive the plant community distribution pattern and species richness in the Arjo Diga forest in western Ethiopia.PLOS ONE

Dear Dr. Berihun,

Thank you for submitting your manuscript to PLOS ONE. After careful consideration, we feel that it has merit but does not fully meet PLOS ONE’s publication criteria as it currently stands. Therefore, we invite you to submit a revised version of the manuscript that addresses the points raised during the review process.

This MS has undergone four rounds of revision but still some of the concerns has not been addressed well. Please look into the comments from one reviewer in this round of review and revise the MS accordingly. Additionally, my earlier comments on reference citation pattern and English language has been addressed partially only. I noticed that there are lots of problems with reference citation pattern and English language (especially in the discussion section). Kindly look them thoroughly, and make a critical revision. Hope authors carefully look into the entire MS, and improve the content in this round of revision.

We look forward to receiving your revised manuscript.

Kind regards,

Bhoj K. Acharya, PhD

Academic Editor

PLOS ONE

Additional Editor Comments:

Reviewers' comments:

Reviewer's Responses to Questions

**Comments to the Author**

1. If the authors have adequately addressed your comments raised in a previous round of review and you feel that this manuscript is now acceptable for publication, you may indicate that here to bypass the “Comments to the Author” section, enter your conflict of interest statement in the “Confidential to Editor” section, and submit your "Accept" recommendation.

Reviewer #4: (No Response)

2. Is the manuscript technically sound, and do the data support the conclusions?

Reviewer #4: Partly

3. Has the statistical analysis been performed appropriately and rigorously? 

Reviewer #4: Yes

4. Have the authors made all data underlying the findings in their manuscript fully available?

Reviewer #4: Yes

5. Is the manuscript presented in an intelligible fashion and written in standard English?

Reviewer #4: No

6. Review Comments to the Author

Reviewer #4: Now submitted the comments, need to wait for the authors response. Objectives need to be clearly defined, further for association studied simple multiple regression could be used. The manuscripts need lots of revision to make the message clear and results needs to be arranged accordingly. Statistical analysis are fine but methodology needs to be elaborated. Data on tree density, volume, and other information could be made available to assess the dominance in a community. There is a lot of scope for improvement and presentation of data in the manuscripts.

7. PLOS authors have the option to publish the peer review history of their article (what does this mean?). If published, this will include your full peer review and any attached files.

Reviewer #4: **Yes: **BHAUSAHEB TAMBAT

---

## [Author Response · Author response to Decision Letter 4]

10 Jul 2024

We have incorporated the comments, questions, and suggestions raised by both the editors and reviewers. Each of these points has been addressed independently in our resubmission.

---

## [Editor Report · Decision Letter 5]

15 Jul 2024

Topography factors and soil variables drive the plant community distribution pattern and species richness in the Arjo Diga forest in western Ethiopia.

PONE-D-23-07720R5

Dear Dr. Berihun,

We’re pleased to inform you that your manuscript has been judged scientifically suitable for publication and will be formally accepted for publication once it meets all outstanding technical requirements.

Kind regards,

Prof. Bhoj K. Acharya

Academic Editor

PLOS ONE

Additional Editor Comments:

The authors have successfully revised the MS now but still there are issues with reference citation which needs to be solved.

For example:

L449: lower than that reported by [51] in the Bonga forest- You should write the name of the authors and then only you have to cite 51 in bracket

L451-452:  A study by[52] revealed that forests subjected: You should write the name of the authors and then only you have to cite 52 in bracket

Such issues are there in introduction, methodology and everywhere in the MS. Please check thoroughly while submitting final files to the journal.
---

## [Editor Report · Acceptance letter]

24 Jul 2024

PONE-D-23-07720R5 

PLOS ONE

Dear Dr. Berihun Tenaw, 

I'm pleased to inform you that your manuscript has been deemed suitable for publication in PLOS ONE. Congratulations! Your manuscript is now being handed over to our production team.

Kind regards, 

on behalf of

Prof. Bhoj K. Acharya 

Academic Editor

PLOS ONE